# Bidirectional Mendelian Randomization Analysis of the Causal Relationship Between Uterine Fibroids and Breast Cancer in East Asian Women

**DOI:** 10.3390/biomedicines13112654

**Published:** 2025-10-29

**Authors:** Young Lee, Je Hyun Seo

**Affiliations:** Veterans Medical Research Institute, Veterans Health Service Medical Center, Seoul 05368, Republic of Korea; lyou7688@gmail.com

**Keywords:** breast cancer, uterine fibroids, estrogen, Mendelian randomization, single-nucleotide polymorphisms

## Abstract

**Background/Objectives**: This study was designed to investigate the potential causal relationship between uterine fibroids (UF) and breast cancer (BC) using genetic data in East Asian populations. **Methods**: We conducted a bidirectional two-sample Mendelian randomization (MR) analysis of UF and BC, selecting exposure-associated single-nucleotide polymorphisms (SNPs) from Biobank Japan and extracting outcome associations from the China Kadoorie Biobank for both directions. The primary estimator was inverse-variance-weighted (IVW), with robustness assessed using the weighted median, MR-Egger regression, and the MR-pleiotropy residual sum and outlier (MR-PRESSO). **Results**: The SNPs with (*p* < 5.0 × 10^−8^) were selected as instrumental variables for UF (*n* = 16) and BC (*n* = 7). There was no evidence of heterogeneity in either direction. Genetically predicted UF was positively associated with BC risk (odds ratio, 1.33; 95% confidence interval, 0.99–1.79; *p* = 0.063), although the association did not reach statistical significance in IVW. In addition, the causal effect of BC on UF was significant (odds ratio, 1.19; 95% confidence interval, 1.08–1.32; *p* < 0.001 in IVW). **Conclusions**: Our study suggested a borderline significant causal effect of UF on BC. Moreover, BC demonstrated a significant causal association with UF, underscoring the need for further research into the role of various mechanisms including estrogen in the relationship between the two diseases.

## 1. Introduction

Uterine fibroids (UF) and breast cancer (BC) are two of the most common hormone-related diseases affecting women worldwide, both contributing substantially to the global burden of women’s health [1,2]. Although these diseases originate in distinct tissues, increasing evidence suggests biological interconnections between them. Both UF and BC are estrogen- and progesterone-dependent tumors, with hormonal exposure playing a central role in their development and progression [3,4,5,6,7,8,9]. However, BC represents a heterogeneous group of tumors with distinct molecular characteristics [10]. According to the Surveillance, Epidemiology, and End Results database (https://seer.cancer.gov/, accessed on 13 October 2025), approximately 10.65% of cases are identified as hormone-independent subtypes, including triple-negative BC. In particular, elevated lifetime estrogen exposure—due to early menarche, late menopause, hormone-replacement therapy, and obesity—has been associated with increased risk for both conditions [3,5,11,12]. Moreover, excess endocrine growth hormone has been reported to be associated with the progression of triple-negative BC [13], suggesting that the potential influence of hormones should be considered.

Additionally, recent observational studies have reported an increased risk of BC among women with a history of UF. A large-scale population-based study using the South Korean National Health Insurance database found that women with symptomatic UF had a significantly elevated risk of developing BC (hazard ratio [HR], 1.30; 95% confidence interval [CI], 1.198–1.41), consistent across all age groups [5]. Similarly, a nationwide analysis from Taiwan reported a more modest but significant risk increase (odds ratio [OR], 1.14; 95% CI, 1.07–1.21) [14]. In the U.S. Sister Study, a 7% increased risk of BC was observed among women with a history of UF, with a notably higher risk among Black women (HR, 1.34; 95% CI, 1.07–1.69) [15].

However, these observational associations are subject to limitations such as residual confounding and reverse causation, which preclude causal inference [16]. Mendelian randomization (MR) has emerged as a powerful approach to address these limitations using germline genetic variants as instrumental variables (IVs) [16,17,18]. Recent MR studies conducted in European populations have suggested a causal effect of genetically predicted UF on increased BC risk, with one study reporting an OR of 1.07 (95% CI, 1.02–1.11; *p* = 0.002) [19] and another reporting an OR of 1.027 (95% CI, 1.006–1.048; *p* = 0.010) [20]. The latter study also evaluated the reverse direction and reported an inverse association between genetically predicted BC and UF risk, although this result was significant only in the MR-Egger analysis and not when using the inverse-variance-weighted (IVW) method (*p* = 0.096) [20]. Since prevalence and clinical manifestations of UF and BC vary substantially across ethnic groups [21,22,23,24,25,26,27], studies on the association of UF and BC in non-European populations are needed. Hence, we conducted a bidirectional two-sample MR analysis to investigate the causal relationship between UF and BC in East Asian individuals using genome-wide association study (GWAS) data from two large biobanks: Biobank Japan (BBJ) and the China Kadoorie Biobank (CKB). Due to the limited numbers of BC and UF cases in CKB, no genome-wide significant single-nucleotide polymorphisms (SNPs) were available from this dataset for instrument selection. Consequently, we used BBJ GWAS data as the exposure data in both directions (UF → BC and BC → UF), with CKB data serving as the outcome data.

## 2. Materials and Methods

### 2.1. Study Design

This study was approved by the Institutional Review Board of the Veterans Health Service Medical Center in Seoul, Korea (approval no. 2025-04-014). Given the retrospective nature of the study and the use of anonymized data, the requirement for informed consent was waived. The study was conducted in accordance with the principles of the Declaration of Helsinki.

### 2.2. Data Sources

We performed a bidirectional two-sample MR analysis to investigate the potential causal relationship between UF and BC. Figure 1 illustrates the schematic diagrams of the analytical study design. Summary-level GWAS statistics for UF and BC were obtained from both BBJ and the CKB. While GWAS data for both traits were available from BBJ and CKB, as mentioned, genome-wide significant SNPs were identified only in the BBJ dataset. Therefore, BBJ GWAS summary statistics were used as the source of exposure data in both directions of the bidirectional MR analysis, and CKB data were used as the outcome dataset. While both BBJ and the CKB have conducted multiple GWAS across a wide range of phenotypes, detailed baseline characteristics of the participants specifically for UF and BC were not provided in the publicly available summary datasets. Therefore, our study relied on the reported sample sizes and case–control distributions from the original GWAS publications [28,29]. For the UF → BC direction, the genetically predicted risk of UF from BBJ (*n* = 80,208; 14,475 cases and 65,733 controls) was tested against the BC risk in CKB (*n* = 45,386; 503 cases and 44,883 controls). In the reverse direction (BC → UF), the BC genetic risk from BBJ (*n* = 79,550; 6325 cases and 73,225 controls) was used to evaluate its effect on UF risk in CKB (*n* = 45,427; 877 cases and 44,550 controls). Detailed information on these datasets is provided in Table 1.

### 2.3. Selection of the Genetic Instrumental Variables

To construct genetic instruments for UF and BC, we selected independent SNPs associated with each trait at genome-wide significance (*p* < 5.0 × 10^−8^) from BBJ. All palindromic SNPs (A/T or G/C) were removed prior to Linkage disequilibrium (LD) clumping because strand information was insufficient to resolve allele orientation, and SNPs absent from either the exposure or outcome GWAS were also excluded. LD clumping was then performed to ensure independence between SNPs, using a window size of 10,000 kb and an LD threshold of *r*^2^ < 0.001 based on East Asian population reference panel from the 1000 Genomes Project Phase 3. Exposure and outcome datasets were harmonized using the harmonise_data function in the TwoSampleMR package to align effect alleles and ensure consistent strand orientation across datasets. To assess instrument strength, we calculated the F-statistic for each SNP and reported the mean F-statistic to evaluate the potential for weak instrument bias. Instruments with F-statistics greater than 10 were considered sufficiently strong to minimize weak instrument bias [30]. When applying the same genome-wide significance threshold to the CKB GWAS datasets, no SNPs passed the criteria, even prior to LD clumping. Therefore, CKB could not be used as the exposure dataset. To avoid weak-instrument bias and to ensure consistency in instrument definition, we used BBJ as the exposure source in both directions.

### 2.4. Mendelian Randomization

The three basic assumptions that underpin MR are as follows: (1) genetic variants that are IVs are strongly linked to the exposure of interest; (2) these variants are not associated with any confounding variables that affect both the exposure and the outcome; and (3) the variants only affect the outcome through the exposure, not through other pathways like horizontal pleiotropy.

### 2.5. Statistical Analysis

The primary analytical method was the IVW method with multiplicative random effects, which provides effective causal estimates when all instruments are valid [30,31,32]. To assess the robustness of causal estimates, we also applied the weighted median method [33] and MR-Egger regression (with and without simulation extrapolation [SIMEX] adjustment) [34,35]. The weighted median method can yield consistent estimates even when up to 50% of the instruments are invalid [33]. MR-Egger regression allows detection and correction of unbalanced horizontal pleiotropy by estimating a non-zero intercept [34], while the SIMEX approach further adjusts for bias when the “no measurement error” (NOME) assumption is violated, particularly when the *I*^2^ statistic is less than 90% [35].

Cochran’s Q statistic in the IVW framework and Rücker’s Q′ statistic under MR-Egger were used to assess between-instrument heterogeneity [31,36]. Significant heterogeneity can be a sign of pleiotropy [31,37]. Furthermore, outlier instruments that could contribute to pleiotropic bias were identified and adjusted using the MR-PRESSO framework [38]. In addition, we assessed statistical power using the mRnd power calculator, available at https://shiny.cnsgenomics.com/mRnd/ (accessed on 13 October 2025). All analyses were performed using the TwoSampleMR (version 0.5.6) and simex (version 1.8) package in R version 3.6.3 (R Core Team, Vienna, Austria). The reporting of this study followed the STROBE-MR checklist [39] to enhance transparency and reproducibility.

## 3. Results

### 3.1. Selection of Instrumental Variables

Sixteen and seven independent SNPs were selected as IVs for UF and BC, respectively. The mean F-statistics for the instruments were 81.51 for UF and 149.34 for BC, indicating that weak instrument bias was unlikely (Table 2). All the F-statistics for SNPs were greater than 10, which means there was a low chance of weak instrument bias. Appendix A provides detailed information about the IVs used in this study. In this MR study, only one SNP (rs6557160 encode *CCDC170;ESR1*) related to estrogen and progesterone reached the GWAS significance level, and therefore only this gene was included. However, MR analysis included *WNT* signaling pathway as an IV, dysregulated in BC, playing a role in cancer cell proliferation, metastasis, stemness, and therapeutic resistance [40]. In addition, fibroblast growth factor receptor 2 signaling, with numerous important functions, including developmental induction, pattern formation, cell growth and differentiation, as well as survival and death, was also highly linked to BC [41].

### 3.2. Heterogeneity and Horizontal Pleiotropy of the Instrumental Variables

We assessed the validity of the IVs by evaluating heterogeneity, horizontal pleiotropy, and the NOME assumption. As shown in Table 2, the *I*^2^ statistic was 88.27% for the UF → BC direction and 76.47% for the reverse direction, indicating a potential violation of the NOME assumption in both analyses (*I*^2^ < 90%). Consequently, MR-Egger with SIMEX adjustment was preferred over standard MR-Egger regression to account for potential measurement error bias. There was no evidence of heterogeneity in either direction (Table 2). Specifically, neither Cochran’s Q test for IVW nor Rücker’s Q′ test for MR-Egger was significant in the analysis of the impact of UF on BC (*p* = 0.338 and *p* = 0.339, respectively) or BC on UF (*p* = 0.978 and *p* = 0.996, respectively), indicating low heterogeneity among the IVs. Horizontal pleiotropy was also assessed using the MR-Egger intercept and MR-PRESSO global test. For UF → BC, the MR-Egger intercepts were −0.045 (*p* = 0.339), and the SIMEX-adjusted intercept was −0.048 (*p* = 0.326), both of which were non-significant, suggesting no evidence of unbalanced pleiotropy. For BC → UF, the MR-Egger intercept was −0.084 (*p* = 0.405), while the SIMEX-adjusted intercept was −0.107 (*p* = 0.011). Although the latter reached statistical significance, it should be interpreted with caution because the MR-PRESSO global test was non-significant (*p* = 0.976). MR-PRESSO global tests were also non-significant in both directions (*p* = 0.388 for UF → BC and *p* = 0.976 for BC → UF), and no outlier SNPs were detected. Taken together, the absence of significant intercepts (except for the SIMEX-adjusted result in BC → UF) and the lack of MR-PRESSO outliers suggest that horizontal pleiotropy is unlikely to materially bias the causal estimates. Based on these findings, the IVW method was considered the most appropriate and robust estimator for both directional analyses [42].

### 3.3. Mendelian Randomization

As mentioned, we conducted a two-sample MR analysis to investigate the potential causal relationship between UF and BC. As seen in Figure 2, a forest plot summarizes the effect estimates derived from various MR methods. In the primary IVW analysis, genetically predicted UF was positively associated with BC risk (OR, 1.33; 95% CI, 0.99–1.79; *p* = 0.063), although the association did not reach statistical significance. The statistical power to detect such an effect was very low (approximately 15%; Appendix A). The weighted median method yielded a significant association (OR, 1.50; 95% CI, 1.01–2.23; *p* = 0.043), while MR-Egger and MR-Egger (SIMEX) showed no significant results. In the reverse direction, as illustrated in Figure 3, genetically predicted BC was significantly associated with an increased risk of UF in the IVW analysis (OR, 1.19; 95% CI, 1.08–1.32; *p* < 0.001). The weighted median estimate was directionally consistent but not significant (OR, 1.15; 95% CI, 0.88–1.50; *p* = 0.312). Although the MR-Egger regression yielded a larger point estimate (OR, 1.90), it was not statistically significant (95% CI, 0.68–5.28; *p* = 0.275). However, after applying SIMEX correction to account for potential bias under the NOME assumption (*I*^2^ = 76.47%), the MR-Egger (SIMEX) estimate was significant (OR, 2.16; 95% CI, 1.60–2.93; *p* = 0.004), reinforcing a potentially causal relationship. The statistical power for the BC → UF analysis was also very low (approximately 10%; Appendix A), suggesting that these findings should be interpreted with caution. The scatterplots showing the SNP–exposure and SNP–outcome associations are presented in Figure 4.

## 4. Discussion

In this study, we found genetic evidence suggesting a potential bidirectional causal relationship between UF and BC in East Asian populations. When UF was evaluated as the exposure, results from the weighted median method indicated a significant positive causal effect on BC risk, while the IVW estimate was borderline significant (*p* = 0.063). Although the MR-Egger methods showed directionally consistent effects, there were wide CIs and lack of significance. Conversely, using BC as the exposure, we observed a significant causal effect on increased UF risk across multiple MR methods, most notably via the IVW (*p* < 0.001) and MR-Egger (SIMEX) (*p* = 0.004) approaches. This suggests a potentially stronger and more consistent effect of BC on UF than vice versa.

Importantly, the prevalence and clinical manifestations of UF and BC vary substantially across ethnic groups [21,22,23,24,25,26,27]. In a large 14-year U.S. cohort study, South Asian, East Asian, and Southeast Asian women had 71%, 47%, and 29% higher UF diagnosis rates, respectively, compared to White women, while Black women had a more than threefold higher rate (incidence rate ratio, 3.11) [23]. Consistently, a cross-sectional study of reproductive-age women revealed the highest UF prevalence rates among Black (35.7%) and Chinese (21.8%) participants compared to White (10.7%) and Hispanic (12.7%) women [25]. Black women also tend to experience earlier onset, larger and more numerous fibroids, and more severe symptoms such as pelvic pain and anemia [22]. Similarly, BC patients exhibit well-documented racial disparities: although White women have the highest overall incidence, Black women are more frequently diagnosed at younger ages and with aggressive subtypes, leading to worse clinical outcomes [27]. Although these incidence differences exist according to ethnicity, our findings are consistent with prior MR studies conducted in European populations, which reported a significant effect of genetically predicted UF on BC risk and suggestive evidence for a reverse association [20]. One previous study on genetic underpinnings showed that UF is associated with BC, especially estrogen receptor-positive BC (OR, 1.54; 95% CI, 1.19–1.99) [43]. In another study, the presence of BC in the baseline assessment of the study population increased the risk of developing UF by 1.5-fold [44]. However, while those studies found an inverse relationship between BC and UF risk, our results show a positive association, warranting further investigation into population-specific genetic architecture and environmental exposures. To further explore population-specific effects within East Asians, we additionally evaluated BC outcome using the KoGES (Korean Genome and Epidemiology Study) dataset [45]. In this supplementary analysis, genetically predicted UF was not significantly associated with BC, and effect directions were inconsistent across MR methods (Appendix A). These results suggest that the causal association may not be robust across all East Asian cohorts and highlight the need for replication in larger Korean datasets.

Many researchers have consistently reported that estrogen and estrogen receptors are the main inducers of UF development [6]. Early menarche and obesity, which are risk factors for BC, are believed to be associated with an increased incidence of UFs [46]. Similarly, estrogen and progesterone exposure are significant risk factors for BC [47]. In addition, recent GWAS results suggested a significant genetic correlation of UF with BC, especially estrogen receptor-positive BC [48]. The possible explanation for the association between UF and BC can be that they share risk factors such as race (Black), age, obesity, environmental pollutants, and endocrine-disrupting chemicals [49,50,51]. In particular, UF-associated genetic variations in *ESR1* genes (Estrogen Receptor 1) encode the estrogen receptor α, which drives the growth of hormone-dependent tumors and influences response to endocrine therapy, and *FSHB* (follicle-stimulating hormone beta subunit) genes encode the beta-subunit of follicle-stimulating hormone. These genes have been linked to cancer, with studies reporting an increased risk for ovarian cancer, endometrial cancer, and BC [52,53]. These SNPs could potentially yield more meaningful results if included in the analysis; however, only *CCDC170;ESR1* (rs6557160) reached significance in the GWAS data used, while the other SNPs did not show genome-wide significant associations (*p* < 5 × 10^−8^), which may represent a limitation of the study. Future studies should focus on MR analyses based on pathologically relevant SNPs.

The current study is one of the first to systematically evaluate the bidirectional causal association between UF and BC in East Asians using genetic instruments. These results contribute novel evidence to help explain the complex interplay between two common hormone-related conditions in women and may have implications for risk stratification and shared pathophysiological pathways. However, this study has several limitations. First, the statistical power of the MR analyses was very low (~15% for UF → BC and ~10% for BC → UF), which substantially limits our ability to detect causal effects. Nevertheless, the BC → UF direction reached statistical significance despite the limited power. These results highlight the importance of replication in larger cohorts to strengthen causal inference. Second, as the analysis was restricted to East Asian participants, the findings may not be generalizable to other ethnic groups. Third, although MR-Egger regression and MR-PRESSO were applied to assess and correct for pleiotropy, the possibility of residual pleiotropy cannot be completely excluded. Another important limitation is the extreme imbalance in case–control numbers in the CKB outcome datasets (e.g., 503 breast cancer cases vs. 44,883 controls). Such imbalance substantially reduces statistical power and precision, potentially leading to wider confidence intervals and unstable effect estimates.

## 5. Conclusions

Our findings indicate a suggestive causal effect of UF on BC and a significant causal association of BC with UF. These bidirectional associations imply that various mechanisms including estrogen may play an important role in the pathogenesis of both diseases, warranting further in-depth investigation.

## Figures and Tables

**Figure 1 biomedicines-13-02654-f001:**
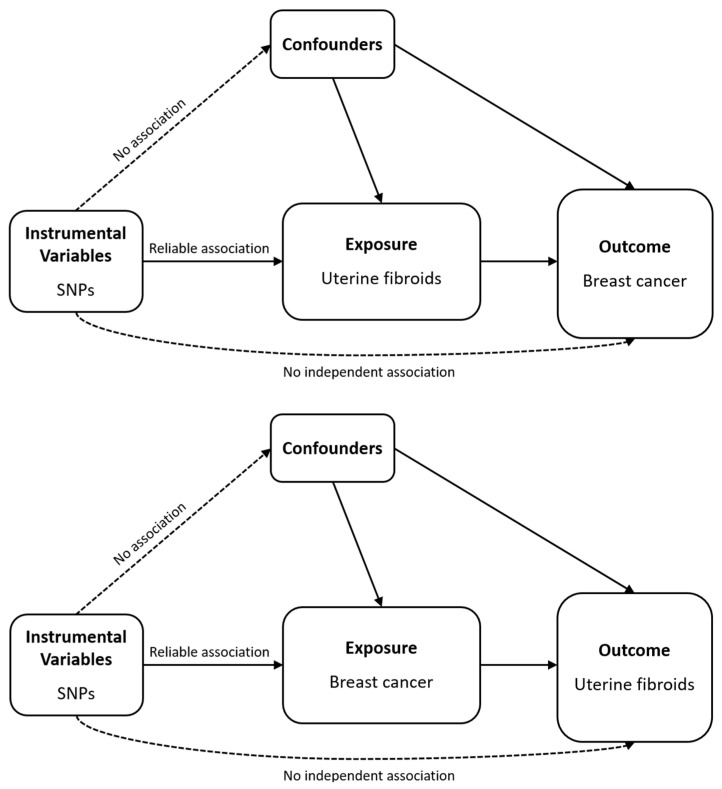
Schematic of the analytical study design. Abbreviations: SNP, single-nucleotide poly-morphism.

**Figure 2 biomedicines-13-02654-f002:**
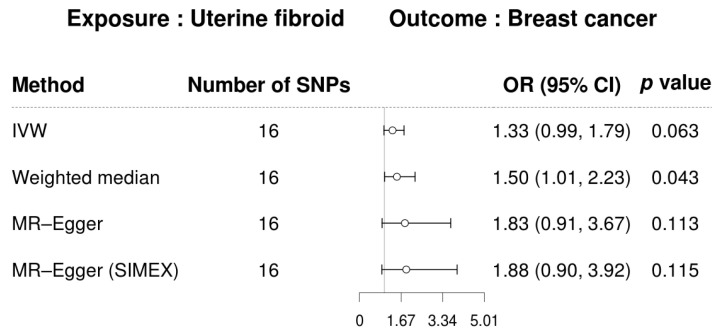
Forest plot of causal associations of uterine fibroids on breast cancer. Abbreviations: CI, confidence interval; IVW, inverse-variance-weighted; MR, Mendelian randomization; OR, odds ratio; SIMEX, simulation extrapolation; SNP, single-nucleotide polymorphism.

**Figure 3 biomedicines-13-02654-f003:**
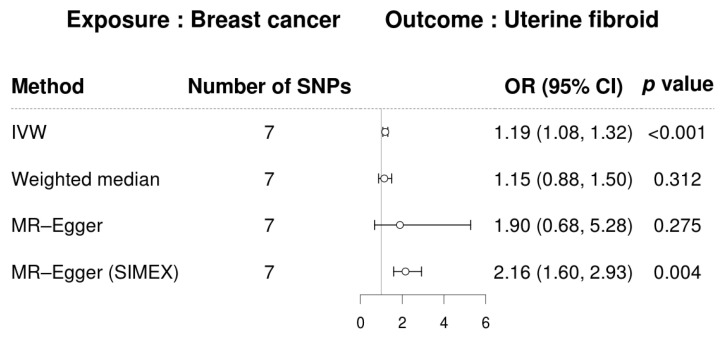
Forest plot of causal associations of breast cancer on uterine fibroids. Abbreviations: CI, confidence interval; IVW, inverse-variance-weighted; MR, Mendelian randomization; OR, odds ratio; SIMEX, simulation extrapolation; SNP, single-nucleotide polymorphism.

**Figure 4 biomedicines-13-02654-f004:**
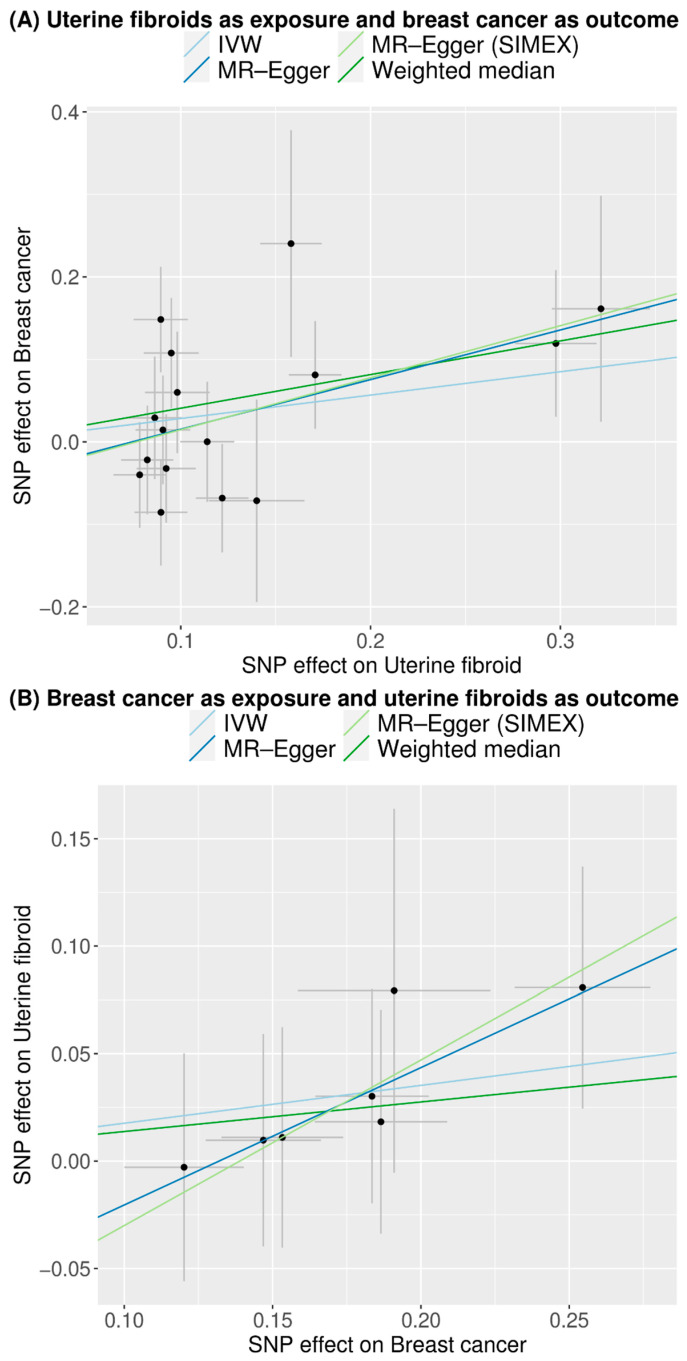
Scatter plots of causal relationship between uterine fibroids and breast cancer. (**A**) Uterine fibroids as exposure and breast cancer as outcome. (**B**) Breast cancer as exposure and uterine fibroids as outcome. Light blue, dark blue, light green, and dark green regression lines represent the IVW, MR-Egger, MR-Egger (SIMEX), and weighted median estimates, respectively. The slope of the line represents the causal effect of each method. Abbreviations: IVW, inverse-variance-weighted; MR, Mendelian randomization; SIMEX, simulation extrapolation; SNP, single-nucleotide polymorphism.

**Table 1 biomedicines-13-02654-t001:** Summary statistics of data sources.

Traits	Data Source	No. of Participants	Population	No. of Variants	URL
Uterine fibroids	BBJ	80,208 (14,475 cases + 65,733 controls)	East Asian	13,401,454	https://pheweb.jp/, accessed on 27 October 2022
Breast cancer	BBJ	79,550 (6325 cases + 73,225 controls)	East Asian	13,401,000	
Uterine fibroids	CKB	45,427 (877 cases + 44,550 controls)	East Asian	8,929,108	https://pheweb.ckbiobank.org/, accessed on 14 January 2025
Breast cancer	CKB	45,386 (503 cases + 44,883 controls)	East Asian	8,578,343	

BBJ, BioBank Japan; CKB, China Kadoorie Biobank.

**Table 2 biomedicines-13-02654-t002:** Heterogeneity and horizontal pleiotropy of instrumental variables.

Exposure	Outcome				Heterogeneity		Horizontal Pleiotropy				
					Cochran’s Q Test from IVW	Rücker’s Q’ Test from MR-Egger	MR-PRESSO Global Test	MR-Egger		MR-Egger (SIMEX)	
		N	*F*	*I*^2^ (%)	*p*	*p*	*p*	Intercept, *β* (SE)	*p*	Intercept, *β* (SE)	*p*
Uterine fibroid	Breast cancer	16	81.51	88.27	0.338	0.339	0.388	−0.045 (0.045)	0.339	−0.048 (0.047)	0.326
Breast cancer	Uterine fibroid	7	149.34	76.47	0.978	0.996	0.976	−0.084 (0.093)	0.405	−0.107 (0.027)	0.011

*β*, beta coefficient; *F*, mean *F*-statistic; IVW, inverse-variance-weighted; MR, Mendelian randomization; N, number of instruments; PRESSO, pleiotropy sum of residuals and outlier; SE, standard error; SIMEX, simulation extrapolation.

## Data Availability

The datasets used and/or analyzed in the current study are available from Biobank Japan (BBJ; https://pheweb.jp/, accessed on 27 October 2022) and China Kadoorie Biobank (CKB; https://pheweb.ckbiobank.org/, accessed on 14 January 2025). The CKB data can be accessed upon application and approval through the CKB PheWeb portal (https://pheweb.ckbiobank.org/about#download_instruction). To comply with Chinese government regulations, the files are encrypted, and decryption keys can be obtained by contacting ckbaccess@ndph.ox.ac.uk with institutional details. The code for the main analyses in this study is available on GitHub (https://github.com/lyou7688/UF_BC_MR, commit 1d51c6d, accessed on 13 October 2025).

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
