# Peer review of "Bidirectional Mendelian Randomization Analysis of the Causal Relationship Between Uterine Fibroids and Breast Cancer in East Asian Women"

_biomedicines, 2025, doi:10.3390/biomedicines13112654_

Round 1
Reviewer 1 Report
Comments and Suggestions for Authors
This manuscript investigates the bidirectional causal relationship between uterine fibroids and breast cancer in East Asian women using Mendelian randomization analysis. The topic is clinically relevant, as both UF and BC are common hormone-related conditions in women, and the exploration of genetic causality provides valuable insights into shared pathophysiological mechanisms. The topic is interesting. However, there are several points that must be revised.
1. Since only Chinese and Japanese populations were included, the results may not fully represent East Asian populations. The authors could consider utilizing Korean data to calculate related results. (https://koges.leelabsg.org/pheno/KoGES_BRCA )
2. Data and code need to be shared either through a code-sharing repo like GitHub or a docker-like system such as codeocean for clear reproducibility of the work.
3. In the Introduction, author mention hormonal exposure playing a central role in BC development and progression. I recommend citing PMID: 39006481 to provide a comprehensive context.
4. MR power calculation is missing. Please add such analysis.
5. Author should use the STROBE-MR checklist to improve the reporting of MR studies and cite PMID: 37198682.
6. A conclusion figure (graphical abstract) will be very useful for the readers.
Author Response
This manuscript investigates the bidirectional causal relationship between uterine fibroids and breast cancer in East Asian women using Mendelian randomization analysis. The topic is clinically relevant, as both UF and BC are common hormone-related conditions in women, and the exploration of genetic causality provides valuable insights into shared pathophysiological mechanisms. The topic is interesting. However, there are several points that must be revised.
Comment 1: Since only Chinese and Japanese populations were included, the results may not fully represent East Asian populations. The authors could consider utilizing Korean data to calculate related results. (https://koges.leelabsg.org/pheno/KoGES_BRCA )
→ We thank the reviewer for this valuable suggestion. Regarding the use of the Korean Genome and Epidemiology Study (KoGES) as an exposure dataset, no SNPs reached genome-wide significance, resulting in zero available instruments. For this reason, KoGES could not be used as the exposure dataset, as doing so would have introduced weak-instrument bias.
In addition to BBJ and CKB, we conducted supplementary analyses using the KoGES breast cancer GWAS as the outcome. When using CKB as the outcome, the IVW analysis suggested a positive direction of effect (OR > 1) but was not statistically significant (p = 0.063). When using KoGES as the outcome, all MR estimates were non-significant; IVW and weighted median showed negative directions, while MR-Egger showed a positive direction. These inconsistent directions and non-significant results suggest limited evidence of a causal effect. We have now included these findings in the Supplementary Material (Additional File S1) and discussed them in the Discussion (page 9, lines 39-44).
“To further explore population-specific effects within East Asians, we additionally eval-uated BC outcome using the KoGES (Korean Genome and Epidemiology Study) dataset [45]. In this supplementary analysis, genetically predicted UF was not significantly as-sociated with BC, and effect directions were inconsistent across MR methods (Addi-tional File S1). These results suggest that the causal association may not be robust across all East Asian cohorts and highlight the need for replication in larger Korean datasets.”
Comment 2 Data and code need to be shared either through a code-sharing repo like GitHub or a docker-like system such as codeocean for clear reproducibility of the work.
→ We agree with the reviewer on the importance of reproducibility. We have deposited core used in this study, including SNP selection, harmonization, and MR analyses, in a publicly accessible GitHub repository (URL: https://github.com/lyou7688/UF_BC_MR). For the datasets, BBJ GWAS summary statistics are publicly available (https://pheweb.jp/, accessed on 30 July 2022). The CKB GWAS summary statistics can be accessed upon application and approval through the CKB PheWeb portal (https://pheweb.ckbiobank.org/about#download_instruction). To comply with Chinese government regulations, the files are encrypted, and decryption keys can be obtained by contacting ckbaccess@ndph.ox.ac.uk with institutional details. Due to data use agreements, these summary statistics cannot be redistributed by us.
Comment 3: In the Introduction, author mention hormonal exposure playing a central role in BC development and progression. I recommend citing PMID: 39006481 to provide a comprehensive context.
→ We agree with the reviewer’s comment to enhance introduction.
(page 2, lines 2-4) “Moreover, excess endocrine growth hormone has been reported to be associated with the progression of triple-negative BC [13] PMID: 39006481, suggesting that the potential influence of hormones should be considered.”
Comment 4. MR power calculation is missing. Please add such analysis.
→ We thank the reviewer for this important comment. We have now performed statistical power calculations using the mRnd power calculator (http://shiny.cnsgenomics.com/mRnd/), based on the proportion of variance explained by the instruments, the sample sizes of the outcome GWAS, and the IVW effect estimates (odds ratios) from our main analyses. The estimated power was approximately 15% for the UF → BC direction and 10% for the BC → UF direction. These results have been added to the Methods (page 4, lines 40-41 to page 5, line 1), Results (page 6, lines 13-15 and 25-27), and the Discussion (page 10, lines 19-23) to acknowledge the limited statistical power of our study.
Comment 5. Author should use the STROBE-MR checklist to improve the reporting of MR studies and cite PMID: 37198682.
→ We fully agree. We revised the manuscript according to the STROBE-MR checklist. We have also cited the STROBE-MR guideline (PMID: 37198682) in the Methods section (page 5, line 3-4).
“The reporting of this study followed the STROBE-MR checklist [39] to enhance trans-parency and reproducibility.”
Comment 6. A conclusion figure (graphical abstract) will be very useful for the readers.
→ We appreciate this constructive suggestion. We have created a new conclusion figure (graphical abstract) summarizing the bidirectional MR study design, datasets, and main findings in a graphical abstract format. This will help readers grasp the study concept and conclusions more intuitively.
Reviewer 2 Report
Comments and Suggestions for Authors
This manuscript addresses the potential bidirectional causal relationship between uterine fibroids (UF) and breast cancer (BC) using a two-sample Mendelian randomization (MR) approach in East Asian populations. The topic is novel, clinically relevant, and timely, as most prior MR studies have focused on European cohorts. The methodological framework is appropriate, with the use of IVW as the main estimator and complementary sensitivity analyses (weighted median, MR-Egger, MR-PRESSO) to assess robustness. The finding of a stronger causal effect of BC on UF than the reverse is intriguing and potentially impactful, given the shared hormonal pathways between these conditions.
While LD clumping parameters are given, the reference panel used for East Asian LD should be explicitly named (e.g., 1000 Genomes East Asian panel).
The outcome datasets (CKB) have very few cases compared to controls (e.g., UF → BC: 503 cases vs. 44,883 controls). This extreme imbalance may reduce statistical power and precision of MR estimates. This limitation should be acknowledged explicitly.
The Methods do not mention allele harmonization between exposure and outcome SNPs, palindromic SNP handling, or strand alignment. These are critical steps in two-sample MR.
Mentioning the exact version of TwoSampleMR and simex packages would improve reproducibility.
While MR-Egger and MR-PRESSO are included, there is no mention of reporting intercept p-values or identifying outlier SNPs. Clarifying how these results were interpreted would improve transparency.
It is unclear why BBJ was used as exposure in both directions rather than using CKB exposure data when available. A brief rationale would strengthen the methodology section.
Author Response
This manuscript addresses the potential bidirectional causal relationship between uterine fibroids (UF) and breast cancer (BC) using a two-sample Mendelian randomization (MR) approach in East Asian populations. The topic is novel, clinically relevant, and timely, as most prior MR studies have focused on European cohorts. The methodological framework is appropriate, with the use of IVW as the main estimator and complementary sensitivity analyses (weighted median, MR-Egger, MR-PRESSO) to assess robustness. The finding of a stronger causal effect of BC on UF than the reverse is intriguing and potentially impactful, given the shared hormonal pathways between these conditions.
Comment 1: While LD clumping parameters are given, the reference panel used for East Asian LD should be explicitly named (e.g., 1000 Genomes East Asian panel).
→ We thank the reviewer for pointing out this important detail. We have now specified the reference panel used for LD clumping in the Methods section. Specifically, we used the East Asian reference panel from the 1000 Genomes Project Phase 3. This information has been added to the Methods (page 4, lines 10-11).
Comment 2: The outcome datasets (CKB) have very few cases compared to controls (e.g., UF → BC: 503 cases vs. 44,883 controls). This extreme imbalance may reduce statistical power and precision of MR estimates. This limitation should be acknowledged explicitly.
→ We appreciate the reviewer’s insightful comment. We agree that the extreme case–control imbalance in the CKB outcome datasets may reduce statistical power and precision of MR estimates. We have now explicitly acknowledged this limitation in the Discussion section (page 10, lines 26-29).
Comment 3: The Methods do not mention allele harmonization between exposure and outcome SNPs, palindromic SNP handling, or strand alignment. These are critical steps in two-sample MR.
→ We thank the reviewer for highlighting this important point. We have revised the Methods section to explicitly describe our harmonization process. Exposure and outcome datasets were harmonized using the harmonise_data function in the TwoSampleMR package to align effect alleles and ensure consistent strand orientation. To avoid strand ambiguity, all palindromic SNPs (A/T or G/C) were removed prior to LD clumping. These details have now been added to the Methods section (page 4, lines 5-8 and 11-13).
Comment 4: Mentioning the exact version of TwoSampleMR and simex packages would improve reproducibility.
→ We agree with the reviewer. We have added the specific versions of the R packages used in our analyses: TwoSampleMR (version 0.5.6) and simex (version 1.8). These details have been added to the Methods section (page 5, line 2) to improve reproducibility.
Comment 5: While MR-Egger and MR-PRESSO are included, there is no mention of reporting intercept p-values or identifying outlier SNPs. Clarifying how these results were interpreted would improve transparency.
→ We appreciate this comment. We have revised the Results section (page 5, lines 35-39 to page 6, lines 1-6) to explicitly report the MR-Egger intercept p-values and to state that no outlier SNPs were identified by MR-PRESSO.
Comment 6: It is unclear why BBJ was used as exposure in both directions rather than using CKB exposure data when available. A brief rationale would strengthen the methodology section.
→ We appreciate the reviewer’s comment. We initially attempted to construct instruments from the CKB exposure GWAS; however, under a genome-wide significance threshold (p < 5×10⁻⁸), no SNPs remained for CKB exposures (i.e., 0 instruments). To avoid weak-instrument bias and to maintain a consistent instrument definition across analyses, we therefore used BBJ as the exposure source in both directions, where genome-wide significant variants were available and had adequate strength. We have added this rationale to the Methods (page 4, lines16-20).
Reviewer 3 Report
Comments and Suggestions for Authors
The manuscript Bidirectional Mendelian Randomization Analysis of the Causal Relationship Between Uterine Fibroids and Breast Cancer in East Asian Women, incorporating bidirectional two-sample MR analysis, interestingly reports the significant causal effect of BC on UF, while the other direction suggests a borderline significant causal effect of UF on BC.
Although the outcome of the research study is very interesting, the manuscript cannot be considered for publication as it stands. The study contains a number of scientific flaws in terms of accuracy of information and lack of information, and their Conclusions need to be revised. Kindly find below my major and minor comments.
- In the Abstract, the authors must define IVs upon first use.
- In the Introduction, the authors cannot state that BC is one of the most common hormone-related diseases, while BC is well known to constitute a heterogeneous group of tumors containing several subtypes, including hormone-dependent and non-hormone-dependent subtypes. This information should be briefly mentioned.
- In the Materials and Methods section, could the authors explain why, among the SNPs selected, there was none selected with genetic variations in the estrogen receptor? Whereas the estrogen/estrogen receptor axis was mentioned as the crucial biological system for the development of both diseases UF and BC.
- In the Results section, more information related to the selected SNPs must be provided, particularly their impact at the cellular level such as inflammation, mesenchymal-epithelial transition (MET), proliferation, migration, etc..
- In the Discussion section, the authors should define or name ESR1 and FSHB for better understanding. In this third paragraph, the authors referred to these two UF-associated genetic variations, which have been reported to link to BC risk. Again, why were these SNPs not used in their study? The use of these SNPs is crucial even for the concept and conclusions of the study.
- In the Conclusions section, the authors must correct this part because there was no evidence involving estrogen unless they generated data after selecting SNPs genetic variants in estrogen receptor gene.
Author Response
The manuscript Bidirectional Mendelian Randomization Analysis of the Causal Relationship Between Uterine Fibroids and Breast Cancer in East Asian Women, incorporating bidirectional two-sample MR analysis, interestingly reports the significant causal effect of BC on UF, while the other direction suggests a borderline significant causal effect of UF on BC.
Although the outcome of the research study is very interesting, the manuscript cannot be considered for publication as it stands. The study contains a number of scientific flaws in terms of accuracy of information and lack of information, and their Conclusions need to be revised. Kindly find below my major and minor comments.
Comment 1: In the Abstract, the authors must define IVs upon first use.
→ We thank the reviewer for this helpful suggestion. Since “instrumental variables” are mentioned only once in the Abstract, we have replaced the abbreviation “IVs” with the full term to improve clarity.
Comment 2: In the Introduction, the authors cannot state that BC is one of the most common hormone-related diseases, while BC is well known to constitute a heterogeneous group of tumors containing several subtypes, including hormone-dependent and non-hormone-dependent subtypes. This information should be briefly mentioned.
→ We appreciate the reviewer’s insightful comment.
Page 1 Lines 6-9 “Although, BC represents a heterogeneous group of tumors with distinct molecular characteristics [10]. According to the Surveillance, Epidemiology, and End Results database (https://seer.cancer.gov/), approximately 10.65% of cases are identified as hormone-independent subtypes, including triple-negative BC.”
Comment 3: In the Materials and Methods section, could the authors explain why, among the SNPs selected, there was none selected with genetic variations in the estrogen receptor? Whereas the estrogen/estrogen receptor axis was mentioned as the crucial biological system for the development of both diseases UF and BC.
→ We appreciate the reviewer’s insightful comment. We added in results section selection IVs.
(Page 5, lines 11-14)
“In this MR study, only one SNP (rs6557160 encode CCDC170;ESR1) related to estrogen and progesterone reached the GWAS significance level, and therefore only this gene was included.”
Comment 4: In the Results section, more information related to the selected SNPs must be provided, particularly their impact at the cellular level such as inflammation, mesenchymal-epithelial transition (MET), proliferation, migration, etc..
→ We appreciate the reviewer’s insightful comment.
(Page 5, lines 14-18)
“However, MR analysis included WNT signaling pathway as IVs, dysregulated in breast cancer, playing a role in cancer cell proliferation, metastasis, stemness, and therapeutic resistance [40]. In addition, fibroblast growth factor receptor 2 signaling numerous important functions, including developmental induction, pattern formation, cell growth and differentiation, as well as survival and death also highly linked BC [41].”
Comment 5: In the Discussion section, the authors should define or name ESR1 and FSHB for better understanding. In this third paragraph, the authors referred to these two UF-associated genetic variations, which have been reported to link to BC risk. Again, why were these SNPs not used in their study? The use of these SNPs is crucial even for the concept and conclusions of the study.
→ Thanks nice suggestions. ESR1 and FSHB explanation enhance understanding.
(Page 10, line 3-13)
“In particular, UF-associated genetic variations in ESR1 genes (Estrogen Receptor 1) encodes the estrogen receptor α, which drives the growth of hormone-dependent tumors and influences response to endocrine therapy and FSHB (follicle-stimulating hormone beta subunit) genes encodes the beta-subunit of follicle-stimulating hormone. These genes have been linked to cancer, with studies reporting an increased risk for ovarian cancer, endometrial cancer, and BC [52,53]. These SNPs could potentially yield more meaningful results if included in the analysis; however, only CCDC170;ESR1 (rs6557160) reached significance in the GWAS data used, while the other SNPs did not show genome-wide significant associations (p < 5×10⁻⁸), which may represent a limitation of the study. Future studies should focus on MR analyses based on pathologically relevant SNPs.”
Comment 6: In the Conclusions section, the authors must correct this part because there was no evidence involving estrogen unless they generated data after selecting SNPs genetic variants in estrogen receptor gene.
→ Thanks nice suggestions. 1 SNPs related estrogen including as IVs, we changed the conclusion accordingly. “These bidirectional associations imply that various mechanisms including estrogen may play an important role in the pathogenesis of both diseases, warranting further in-depth investigation.”
Reviewer 4 Report
Comments and Suggestions for Authors
The presented paper is devoted to the analysis of the relationship between single nucleotide polymorphisms in uterine fibrosis and breast cancer. The paper seems to be relevant to the scope of the journal, and the amount of the presented data is enough for the research paper, however, there are several issues to be addressed.
- The subsection “Patients” with the description of the cohorts of the patients with the clinical and demographic data should be added to the section Methods.
- The criteria of inclusion and exclusion should be provided.
- the statistical analysis should be described separately in the section Methods (subsection “Statistical analysis”).
- The description of SNPs revealed in the analysis should be added to the main text from the supplement. The biological meaning of SNPs found should be briefly described in the text. The possible role of the specific SNPs in the progression of both diseases should be discussed.
- Abbreviation list is missing.
- Text formatting should be unified according to the journal rules.
- The extensive English editing should be performed: typos, long sentences are difficult to read and must be corrected. The text should be revised to be more concise, to be shortened and better organized.
The extensive English editing should be performed: typos, long sentences are difficult to read and must be corrected. The text should be revised to be more concise, to be shortened and better organized.
Author Response
The presented paper is devoted to the analysis of the relationship between single nucleotide polymorphisms in uterine fibrosis and breast cancer. The paper seems to be relevant to the scope of the journal, and the amount of the presented data is enough for the research paper, however, there are several issues to be addressed.
Comment 1: The subsection “Patients” with the description of the cohorts of the patients with the clinical and demographic data should be added to the section Methods.
→ We thank the reviewer for this comment. As this study is a two-sample MR analysis, we relied on publicly available summary-level GWAS data from BBJ and the CKB. While both BBJ and CKB have conducted multiple GWAS across a wide range of phenotypes, detailed baseline characteristics of the participants specifically for UF and BC were not provided in the publicly available summary datasets. In the Data sources section, we have described the contributing cohorts and reported the number of cases and controls for each phenotype based on the original GWAS publications. We have also added an explicit statement noting the absence of detailed demographic and clinical characteristics for these phenotypes, to clarify why a separate “Patients” subsection could not be provided (page 2, line 49 to page 3, lines 1-4).
Comment 2: The criteria of inclusion and exclusion should be provided.
→ We appreciate the reviewer’s suggestion. Because this is a two-sample MR study based on summary-level GWAS data, inclusion and exclusion criteria are not applicable to individual patients. Instead, these criteria apply to the selection of SNPs as instrumental variables. We have revised the Methods (page 4, lines 5-13) to clearly describe our instrument selection procedure: genome-wide significance threshold (p < 5×10⁻⁸), LD clumping parameters (r² < 0.001, 10,000 kb window), allele harmonization between exposure and outcome datasets, exclusion of SNPs not available in both datasets, and removal of all palindromic SNPs (A/T or G/C) because strand information was insufficient to resolve allele orientation.
Comment 3: the statistical analysis should be described separately in the section Methods (subsection “Statistical analysis”).
→ We thank the reviewer for this helpful suggestion. In the revised manuscript, we have reorganized the Methods section to present “Mendelian Randomization” (Section 2.4) and “Statistical Analysis” (Section 2.5) as separate subsections. Section 2.4 now describes the fundamental MR assumptions, while Section 2.5 provides a detailed description of the statistical methods used, including IVW, weighted median, MR-Egger regression with and without SIMEX adjustment, heterogeneity tests, MR-PRESSO, and software packages. The content has not been changed, but this reorganization improves readability and clarity.
Comment 4: The description of SNPs revealed in the analysis should be added to the main text from the supplement. The biological meaning of SNPs found should be briefly described in the text. The possible role of the specific SNPs in the progression of both diseases should be discussed.
→ We added in results section (page 5, lines 11-18).
“In this MR study, only one SNP (rs6557160 encode CCDC170;ESR1) related to estrogen and progesterone reached the GWAS significance level, and therefore only this gene was included.”
“However, MR analysis included WNT signaling pathway as IVs, dysregulated in breast cancer, playing a role in cancer cell proliferation, metastasis, stemness, and therapeutic resistance [40]. In addition, fibroblast growth factor receptor 2 signaling numerous important functions, including developmental induction, pattern formation, cell growth and differentiation, as well as survival and death also highly linked BC [41].”
Comment 5: Abbreviation list is missing.
→ We appreciate this suggestion. We have added a complete list of abbreviations at the end of the manuscript (page 11).
Comment 6: Text formatting should be unified according to the journal rules.
→ We thank the reviewer for pointing this out. We have revised the entire manuscript to unify text formatting, including section headings, figure/table captions, reference style, and abbreviations, in accordance with the journal’s formatting requirements.
Comment 7: The extensive English editing should be performed: typos, long sentences are difficult to read and must be corrected. The text should be revised to be more concise, to be shortened and better organized.
→ We appreciate the reviewer’s observation. The manuscript has undergone extensive English editing by a professional editor. We have corrected typographical errors, shortened lengthy sentences, and reorganized the text to improve clarity, readability, and conciseness.
Round 2
Reviewer 1 Report
Comments and Suggestions for Authors
well revision
Reviewer 2 Report
Comments and Suggestions for Authors
The authors have thoroughly addressed all previous comments with clear revisions.
The methodological clarifications (LD reference panel, allele harmonization, software versions) enhance reproducibility.
Reporting of MR-Egger intercept p-values and MR-PRESSO outcomes improves transparency.
The manuscript is now methodologically sound, clearly written, and ready for publication.
Reviewer 3 Report
Comments and Suggestions for Authors
I would like to thank the authors for taking my comments into consideration, which improved the quality of the paper.
Reviewer 4 Report
Comments and Suggestions for Authors
No further comments